# Gastric Mucosal Protective Effects of *Cinnamomum cassia* in a Rat Model of Ethanol-Induced Gastric Injury

**DOI:** 10.3390/nu16010055

**Published:** 2023-12-23

**Authors:** Young-Min Han, Moon-Young Song, Da-Young Lee, Seung-Won Lee, Hye-Rin Ahn, Jihee Yoo, Hyo Jun Kim, Eun-Hee Kim

**Affiliations:** 1College of Pharmacy and Institute of Pharmaceutical Sciences, CHA University, Seongnam 13488, Republic of Korea; han.ymin3@gmail.com (Y.-M.H.); wso219@naver.com (M.-Y.S.); angela8804@naver.com (D.-Y.L.); seungwon1194@naver.com (S.-W.L.); ahyerin0627@kakao.com (H.-R.A.); 2CHlabs Corporation, Seoul 07249, Republic of Korea; jhyoo@chlabs.co.kr; 3Chong Kun Dang Healthcare, Seoul 07249, Republic of Korea; kimxg@ckdhc.com

**Keywords:** gastritis, gastric ulcer, *Cinnamomum cassia*, ethanol, gastric mucosal defense

## Abstract

*Cinnamomum cassia* (cassia) is a tropical aromatic evergreen tree of the Lauraceae family well known for its fragrance and spicy flavor and widely used in Asian traditional medicine. It has recently garnered attention for its diverse potential health benefits, including anti-inflammatory, anti-cancer, and anti-diabetic properties. However, the gastroprotective effect of *C. cassia*, particularly against ethanol-induced gastric damage, remains unclear. We investigated the potential gastroprotective property of *C. cassia* and the underlying mechanisms of action in a rat model of ethanol-induced gastric injury. To assess its effectiveness, rats were fed *C. cassia* for a 14-day period prior to inducing gastric damage by oral administration of ethanol. Our results indicated that pre-treatment with *C. cassia* mitigated ethanol-induced gastric mucosal lesions and bleeding. Reduced gastric acid secretion and expression of acid secretion-linked receptors were also observed. Additionally, pretreatment with *C. cassia* led to decreased levels of inflammatory factors, including TNF-α, p-p65, and IκBα. Notably, *C. cassia* upregulated the expressions of HO1 and HSP90, with particular emphasis on the enhanced expression of PAS and MUC, the crucial gastric mucosa defense molecules. These findings suggest that *C. cassia* has protective effects on the gastric mucosa and can effectively reduce oxidative stress and inflammation.

## 1. Introduction

Gastritis is a common ailment that afflicts a large number of individuals worldwide each year [1]. It is an inflammatory condition that affects the gastrointestinal mucosa, and if it remains untreated, can progress to chronic gastritis, gastric ulcers, and, in certain instances, even gastric cancer [1]. Several factors influence damage to the gastrointestinal mucosa including eating habits, smoking, severe stress, and heavy drinking, as well as Helicobacter pylori infections and use of non-steroidal anti-inflammatory drugs (NSAID). Among these, ethanol is known to contribute to the severity of gastric mucosal injury by disrupting the mucosal integrity, gastric mucosal bleeding, and mucosal cell death [2]. Gastric mucosal damage induced by neutrophil infiltration into gastric tissues is closely associated with elevated levels of pro-inflammatory cytokines [3]. Oxidative stress plays a significant role in the pathogenesis of alcohol-induced gastritis [4]. Reactive oxygen species (ROS) generated by activated white blood cells contribute to mucosal injury by depleting glutathione, reducing the activity of glutathione peroxidase, and compromising antioxidant defenses, such as total antioxidant capacity [5]. Excessive gastric acid secretion can exacerbate gastritis, with the histamine 2 receptor (H2R) binding to histamine, triggering the release of gastric acid into the gastric lumen [6]. Consequently, medications prescribed for gastric ulcers include proton pump inhibitors (PPI), and H2R antagonists that reduce gastric acid secretion and enhance mucosal protection [7]. However, despite their clinical effectiveness, these drugs are associated with several side effects [7]. While these drugs exhibit no adverse effects during a short 2-week treatment duration, prolonged use has been associated with complications such as infection, kidney disease, hypergastrinemia, nutrient malabsorption, and dementia [8,9,10]. Clinical meta-analyses have demonstrated that long-term PPI use can exacerbate various conditions, including gastric cancer [11]. It can lead to hypergastrinemia due to potent gastric acid suppression, triggering intestinal chromaffin cell proliferation [12]. Prolonged PPI use, particularly in the presence of *H. pylori* infection, has been linked to the development of gastric cancer [12]. Consequently, exploring alternatives to mitigate these treatment side effects becomes imperative.

Intriguingly, recent research shows that natural products and plant extracts have lower toxicity in animal trials than synthetic pharmaceuticals, suggesting a potential remedy to the limitations of present therapies [13]. Recently, plant-derived phytochemicals have emerged as potential drugs for treatment of gastritis, and their usage has shown no toxicological risk even with long-term chronic exposure [13]. Among them, those from *Cinnamomum cassia* (cassia) have shown promise for treatment and protection against gastritis. Toxicological data on the pharmacokinetics, clinical trials, and safety of *C. cassia* support its use as an alternate medication without side effects, addressing the limitations posed by current synthetic drugs [14,15]. Recent studies have shown that anti-cancer chemotherapy using natural products has fewer side effects than conventional therapeutics [16]. Plant-derived phytochemicals include secondary metabolites such as alkaloids, flavonoids, and quinones, and these components have been reported to be effective in treating various diseases [17]. In addition, dandelion root extract has been reported to inhibit the proliferation and migration of gastric cancer cells, and consumption of citrus fruits has been reported to reduce gastric cancer risk [16,18]. Moreover, plant extracts rich in polyphenols have been reported to have a beneficial effect on the gastric intestinal tracts, such as in inflammatory bowel diseases [17]. Therefore, exploring new natural products can be a promising therapeutic strategy for treating gastritis and gastric ulcers.

*C. cassia*, an evergreen tree commonly utilized in Asian traditional medicine and as a culinary ingredient is noteworthy [19]. In-depth analysis reveals that *C. cassia* boasts a diverse array of constituents, exceeding 160 in total [19], with terpenoids being the most abundant phytochemical, underscoring their significance [19]. Furthermore, phenylpropanoids have been identified as noteworthy bioactive components contributing to the plant’s pharmacological profile [19]. Its intricate composition extends to encompass glycosides, lignans, and lactones, as corroborated by recent research findings [19]. These compounds have been the subject of various studies highlighting its potential anti-tumor, anti-inflammatory, anti-diabetic, anti-obesity, antiviral, antibacterial, cardio-vascular-protective, cytoprotective, and neuroprotective effects [19]. Particularly, studies conducted to investigate the effects of orally administered *C. cassia* on gastric mucosal damage in gastrointestinal disorders have shown positive outcomes indicating its anti-inflammatory effects in conditions such as *H. pylori* infection and NSAID usage [20,21]. Furthermore, several studies have discussed both the mitigation of gastric mucosal damage and accompanying benefits of nontoxic natural products in animal models [22]. Hence, in this study, we used a rat model to thoroughly investigate the impact of *C. cassia* on ethanol-induced acute gastric mucosal damage, with a primary focus on unraveling the intricate underlying mechanisms.

## 2. Materials and Methods

### 2.1. Preparation of Cinnamon Extracts

*C. cassia* used in the experiments was obtained from CHLabs Co., Ltd. (Seoul, Republic of Korea). *C. cassia* was extracted twice with distilled water for 5 h at 100 °C. The extract was filtered, concentrated, and spray dried using dextrin.

### 2.2. Cinnamic Acid Analysis by High-Performance Liquid Chromatography (HPLC)

HPLC was performed using a Waters E2695 Separations Module HPLC system (Milford, MA, USA) for detection and extraction of cinnamic acid from *C. cassia* extracts. A PDA detector (Waters) was used for detection of the components and the separation was carried out with a Phenomenex Gemini C18 column (250 × 4.6 mm, 5 μm) (Waters). The *C. cassia* extract was dissolved with 50% methanol, and 10 μL of it was injected into the column. Cinnamic acid standard for HPLC analysis was obtained from Sigma–Aldrich (St. Louis, MO, USA). To detect cinnamic acid, the mobile phase was composed of distilled water with 0.1% (*v/v*) trifluoro acetic acid (TFA) (solvent A) and acetonitrile (ACN) with 0.1% (*v/v*) TFA (solvent B) (Figure 1). The gradient program was 0–30 min, 10–90% B; 30–35 min, 90% B; 35–40 min, 90–10% B; 40–50 min, 10% B at a flow rate of 0.7 mL/min at 280 nm.

### 2.3. Animal Experiments

Sprague-Dawley (SD) rats (male, 5 weeks old) were procured from Orient Biology (Seoul, Republic of Korea). All animals were kept in cages with humidity levels controlled automatically, subjected to a 12 h light/dark cycle at a temperature of 24 °C, and provided ad libitum access to water and a rodent chow diet. The animals were cared for in compliance with the guidelines of the Institutional Animal Care and Use Committee (IACUC) at the CHA University Animal Center (reference number IACUC230081). The rats were acclimatized for 1 week prior to the experiment. Subsequently, the rats were divided into five groups (*n* = 10 each): saline (NC), ethanol treatment (E), ethanol with 20 mg/kg cinnamon extract (CL), ethanol with 30 mg/kg cinnamon extract (CH), and ethanol with 30 mg/kg rebamipide (PC). The experimental groups are described in Table 1. Rebamipide or cinnamon extract (1 mL) was orally administered to the rats dissolved in distilled water for 14 days. At the end of the 14-day period, acute gastric damage was induced by oral administration 1 mL of 100% ethanol to all groups except the NC group. Subsequently, the rats were euthanized 2 h after ethanol administration, and gastric juice, stomach tissue, and serum samples were collected for analysis.

### 2.4. Measurement of Gastric Juice PH and Total Acidity

After dissection, gastric secretions were collected in graduated microcentrifuge tubes using a syringe. The collected gastric juice was centrifuged at 2000 rpm for 10 min. From the resulting supernatant, 1 mL aliquots of gastric juice were diluted with an equal volume of distilled water, and the pH of the solution was determined using a pH meter. 

To determine the total acidity, 1 mL of diluted gastric juice (diluted with 1 mL of dis-tilled water) was transferred into a 50 mL conical tube. To this solution, 2–3 drops of 3-(dimethylamino)benzyl alcohol (Sigma Aldrich, St. Louis, MO, USA) was added as an indicator, and titration was carried out using 0.1 N NaOH until a fuchsia pink color appeared. The volume of 0.1 N NaOH consumed during titration was recorded. Total acidity was calculated using the following formula: total acidity (mEq/L) = (volume of NaOH × normality)/volume of gastric juice (L).

### 2.5. Gross Lesions Index

After euthanasia, the isolated stomachs were carefully opened and washed with ice-cold saline. To evaluate the extent of gross mucosal pathology, photographs of the mucosal surfaces of the stomach were captured using a digital camera. Gross damage to the gastric mucosa was assessed using a gross ulcer index. The severity of gross ulcers was quantified using a scoring system, and the mean gross lesion score was expressed as the ulcer index. The ulcers were scored as follows: normal stomach (0), red coloration (0.5), spot ulcers (1), hemorrhagic streaks (1.5), <5 ulcers (2), and >5 ulcers (3).

### 2.6. Histological Analysis

For histopathological analysis, the stomachs were fixed with 10% formalin and processed using a standard method, followed by embedding them in paraffin. Sections of 4 μm thickness were then prepared and stained with hematoxylin and eosin (H&E) following the protocol [23]. To assess the pathological changes resulting from ethanol-induced damage, including inflammatory infiltration, erosive lesions, hemorrhage, and ulceration, three gastroenterologists, who were blinded to the experimental groups, performed grading using a previously defined histological injury index [24]. Inflammation was graded based on the infiltration of inflammatory cells as follows: absent (0), infiltration under the lamina propria (1), infiltration involving half of the mucosa (2), and infiltration extending into the epithelial gland layer throughout the mucosa (3). Erosion was graded according to the proportion of erosive lesions as follows: no erosion (0), loss of the epithelial gland layer (1), erosion affecting two-thirds of the mucosa (2), and erosion involving the entire mucosa (3).

### 2.7. Measurement of Nitric Oxide (NO), Myeloperoxidase (MPO), Tumor Necrosis Factor-Alpha (TNF-α) and Interleukin (IL)-1β

Following euthanasia, blood samples were collected for enzyme-linked immunosorbent assay (ELISA). After centrifugation, the levels of NO (iNtron Biotechnology, Seongnam, Republic of Korea), MPO (Abcam, Cambridge, United Kingdom), TNF-α (R&D systems, Minneapolis, MN, USA), and IL-1β (R&D systems, MN, USA) in the supernatant were quantified using ELISA. These procedures were performed in strict accordance with manufacturer’s instructions. 

### 2.8. Immunohistochemical (IHC) Staining for F4/80 and Periodic Acid and Schiff’s (PAS) Staining

For IHC assessment, 4 μm paraffin-embedded stomach sections were mounted on glass slides for the detection of the target proteins. PAS staining, a histochemical method for detecting glycoconjugates, was performed according to the procedure described by Pandurangan [25]. This involved the use of a 2% PAS reagent, with incubation carried out in the dark for 20 min. 

For F4/80 antibody staining, antigen retrieval, blocking of endogenous peroxidases, and mitigation of nonspecific protein binding were performed. Slide sections were initially incubated with primary antibodies specific to F4/80 (Thermo Fisher Scientific, Hampton, NH, USA), followed by incubation with horseradish peroxidase-conjugated secondary antibodies. Subsequently, a chromogen was used for color development. 

All stained slides were developed with 3,3′ diaminobenzidine followed by counterstaining with hematoxylin. The percentage of cells positively stained for the target protein was estimated and the following scoring system was applied: cases with ≤5% positively stained cells were scored as 1; 5–20%, 2; 20–50%, 3; 50–80%, 4; and ≥80% stained cells were scored as 5.

### 2.9. Protein Extraction and Immunoblotting

The stomach tissues were homogenized in ice-cold cell lysis buffer (Cell Signaling Technology, Danvers, MA, USA) supplemented with phosphatase and protease inhibitors (Roche Applied Science, Penzberg, Germany). Following homogenization, the samples were centrifuged and the resulting supernatants were collected. Immunoblotting was performed according to a previously published protocol [26]. A detailed list of the antibodies is provided in Table 2.

### 2.10. mRNA Extraction and Real-Time Quantitative Reverse Transcription PCR (qRT-PCR)

This assay was conducted following the procedures outlined in a previous publication [26]. Briefly, total RNA was extracted from cells using TRIzol reagent (Invitrogen, Carlsbad, CA, USA), in accordance with the manufacturer’s instructions. Reverse transcription was carried out using 2 μg of pure RNA with SuperScript II reverse transcriptase (Invitrogen, CA, USA). The expression levels of specific genes were determined by qRT-PCR using a ViiATM 7 Real-Time PCR System (Applied Biosystems, Waltham, MA, USA). All the oligonucleotide primers listed in Table 3 were synthesized by Macrogen (Seoul, Republic of Korea).

### 2.11. Statistical Analysis

Results are expressed as the mean ± standard deviation (SD). Statistical analyses were performed using GraphPad software 5 (GraphPad Software, La Jolla, CA, USA). Comparisons between means were performed using one-way analysis of variance (ANOVA), and statistical significance between groups was determined using Tukey’s multiple comparison test. Statistical significance was set at *p* < 0.05.

## 3. Results

### 3.1. C. cassia Pretreament Protects against Ethanol-Induced Gastric Damage in Rats

In this study, we investigated the gastroprotective potential of *C. cassia* in a rat model of ethanol-induced gastric damage. Compared with the saline-treated control group, the ethanol-treated group exhibited notable structural pathological changes, including hemorrhagic lesions and disruption of epithelial cells in the gastric mucosa (Figure 2A). Development of multiple hemorrhagic lesions in the gastric mucosa as a results of ethanol administration resulted in an increased lesion index (Figure 2B). However, in the groups treated with *C. cassia* or rebamipide, there was a notable inhibitory effect on gastric ulcer lesions, leading to a significant reduction in severity (Figure 2B). Histological analysis further revealed that compared to the normal control group, the ethanol-treated group displayed pathological characteristics such as epithelial cell loss, swelling, bleeding, and infiltration of inflammatory cells (Figure 2C). Conversely, the groups to whom *C. cassia* was administered demonstrated effective amelioration of gastric damage (Figure 2B). The total histological score, which increased in the ethanol group, was significantly reduced by the administration of *C. cassia* (Figure 2D). Notably, high concentrations of *C. cassia* exhibited potent protective effects (Figure 2D). The observation indicated that the low concentration of *C. cassia* and rebamipide exhibited comparable levels of pathological characteristics (Figure 2C,D). Collectively, these findings demonstrate that administration of *C. cassia* can significantly reduce the ulcer index and histopathological score in ethanol-induced gastric damage.

### 3.2. C. cassia Affects Gastric Acid Secretion and Expression of Gastric Acid Secretion-Related Receptors in Ethanol-Induced Gastric Damage Rats

The acidity of the gastric juice, which can influence the damage to gastrointestinal mucosa, was measured along with the determination of gene expression changes related to gastric juice secretion via qRT-PCR analysis. In the ethanol-treated group, the pH of the gastric acid was significantly lower than that in the control group, whereas the total acidity was higher (190% compared to N group) (Figure 3A,B). However, following administration of *C. cassia* or rebamipide, there was a notable increase in gastric acid pH and a significant reduction in total acidity (about 70% compared to E group) (Figure 3A,B). Moreover, the mRNA expression levels of H2R, cholecystokinin 2 receptor (CCK2R), and muscarine acetylcholine receptor (M3R) in the stomachs of rats treated with ethanol were higher than those in rats treated with saline (Figure 3C–E). Conversely, compared with the ethanol-treated group, the rats in the groups treated with *C. cassia* exhibited lower mRNA expression levels, with the rebamipide-treated group exhibiting the most significant reduction (Figure 3C–E).

### 3.3. C. cassia Downregulates Pro-Inflammatory Signaling Pathway in Ethanol-Induced Gastric Damage in Rats

Total RNA was extracted using a commercial kit, and mRNA levels of inducible nitric oxide synthase (iNOS), IL-6, and TNF-α were quantified via qRT-PCR to evaluate the impact of *C. cassia* on the transcription of inflammatory mediators in the ethanol-induced gastric damage model. The secretion of TNF-α and IL-1β, which increased due to ethanol exposure (184% and 127% compared to N group, respectively), was significantly reduced by treatment with *C. cassia* or rebamipide (about 55% and 80% compared to E group, respectively), (Figure 4A, B). Additionally, the ethanol-treated group exhibited a substantial up-regulation in the expression of iNOS, IL-6, and TNF-α mRNA when compared with that in the normal control group (Figure 4C). Conversely, the administration of *C. cassia* in a dose-dependent manner led to a down-regulation of iNOS, IL-6, and TNF-α expression, with the most pronounced reduction observed in the group treated with rebamipide (Figure 4C). Furthermore, the protein expression levels of p-IκBα, p-p65, and p-STAT3 in the stomach of ethanol-treated rats were significantly higher than those in the control group rats (Figure 4D). 

However, within the *C. cassia* supplementation groups in comparison to the E group, the protein expression levels of p-IκBα, p-p65, and p-STAT3 were notably diminished, with the most substantial reductions observed in the high-concentration *C. cassia* group and the rebamipide administration group (Figure 4D). Given that the macrophages and monocytes are the sources of these inflammatory cytokines, we investigated the macrophages using F4/80 IHC staining. As shown in Figure 4E, F4/80 expression significantly increased following ethanol exposure, but underwent a notable reduction in the *C. cassia*-administered groups which was more effective than the decrease in the rebamipide-treated group (Figure 4E). Collectively, these observations indicated that treatment with *C. cassia* exerted anti-inflammatory effects against ethanol-induced gastric damage.

### 3.4. C. cassia Upregulates Antioxidant and Mucosal Defense-Associated Genes in Ethanol-Induced Gastric Damage Rats

We examined the oxidative stress markers, including NO and MPO, to validate the antioxidant properties of *C. cassia* extract. Our results confirmed a significant reduction in the levels of NO and MPO in the *C. cassia* administration group compared to the ethanol administration group (about 80% and 58% compared to E group, respectively) (Figure 5A,B). Furthermore, we examined the expression of HO-1 and heat shock proteins (HSP)-90 as indicators of antioxidative mechanisms. Specifically, the group to whom a high concentration of *C. cassia* was administered exhibited a significant increase in HSP90 and HO-1 expression (Figure 5C). We also investigated the mRNA expression of MUC5A and MUC6, which are crucial components of the gastric mucosa. Remarkably, administration of a high concentration of *C. cassia* substantially upregulated the expression of both MUC5A and MUC6 (Figure 5D). To assess whether *C. cassia* protects against ethanol-induced gastric damage, we performed PAS staining of gastric tissues. As shown in Figure 5F, administration of *C. cassia* significantly preserved the number of PAS-positive gastric glands, even in the presence of ethanol exposure (Figure 5F). Interestingly, the positive control, rebamipide, used for the antioxidant- and defense mechanism-related factors in Figure 5A,B,E, did not have significant effects. This underscores that while *C. cassia* shares anti-inflammatory properties with rebamipide, a commonly used drug for gastritis treatment, it specializes in antioxidant effects and the reinforcement of gastric mucosa defense mechanisms.

## 4. Discussion 

Medicinal plants and their bioactive constituents have been extensively used in traditional medicine to treat gastrointestinal ailments. Numerous studies have documented the anti-ulcerative properties of these plants and their active compounds through animal experimentations [27,28]. In this study, we investigated the gastroprotective effects of *C. cassia* against acute hemorrhagic gastric injury induced by absolute alcohol administration in SD rats. 

Ethanol expedites gastric damage by promoting the infiltration of immune cells, thus initiating inflammatory cascades [29]. Neutrophil infiltration into the gastric mucosa plays a pivotal role in the pathogenesis and damage caused by gastric mucosal inflammation and damage [30]. Migrating neutrophils and macrophages release numerous pro-inflammatory cytokines and ROS during gastric inflammation [31]. The findings of our study indicate that the levels of NO, MPO, and pro-inflammatory cytokines associated with ROS and inflammation were elevated in the ethanol-treated group. However, most *C. cassia* extracts significantly mitigated these effects.

NO is recognized for its regulatory role in gastric acid secretion in animals, and several studies have indicated elevated NO levels in gastritis [32]. Furthermore, it has been reported that the activity of iNOS increases in patients infected with *H. pylori* [33]. Furthermore, ethanol administration has been shown to upregulate the production of pro-inflammatory cytokines, such as TNF-α, known to mediate the NF-κB signaling pathway [34]. The translocation of NF-κB to the nucleus induces transcriptional activation of pro-inflammatory cytokines [35]. Among these, the IL-6 induced activation of STAT3 signaling is noteworthy [35]. Several studies underscore the significance of regulating NF-κB as a therapeutic target for gastric damage [36]. Additionally, given the interplay between STAT3 and NF-κB, targeting both transcription factors represents a crucial avenue for treating inflammation-mediated diseases [35]. In our study, we observed that ethanol administration significantly augmented the phosphorylation of STAT3, cytokines, and NF-κB. However, treatment with *C. cassia* mitigated these changes, suggesting that *C. cassia* may possess anti-inflammatory properties through the modulation of both NF-κB and STAT3 signaling pathways. 

Studies conducted by other investigators with cinnamon extract have revealed that in an EtOH/HCl-induced gastric mucosal injury model, cinnamon extract inhibits inflammation-related genes and signaling pathways by targeting the NF-κB inflammatory signaling pathway, consequently inducing anti-gastritis effects [21]. Additionally, the application of cinnamon extract in RAW cells has been demonstrated to attenuate ROS accumulation generated post-lipopolysaccharide stimulation, mitigating the phosphorylation of transcription factors such as NF-κB and STAT3, thereby exerting anti-inflammatory effects [20]. Furthermore, administration of cinnamon extract in an acute gastric injury animal model induced by indomethacin has been reported to increase gastric juice volume and reduce inflammatory mechanisms involving NF-κB [20]. These findings align with our research. However, as the gastric protective effect of cinnamon extract extends beyond a simple explanation of anti-inflammatory and antioxidant properties, our study is focused on uncovering additional mechanisms in this regard.

Oxidative stress is known to cause structural damage to the gastric mucosa, resulting in the loss of mucosal integrity due to detrimental factors, such as ethanol [37]. The gastric mucus plays a pivotal role in epithelial defense [37]. Our study revealed a significant decrease in the expression of PAS, a marker of gastric mucus, in the ethanol-treated group with severe gastric damage. However, it was evident that the gastric epithelium was better preserved in the group to whom *C. cassia was* administered. This suggests that the administration of *C. cassia* may confer protective effects. This observation is consistent with previous research emphasizing the antioxidant properties of *C. cassia*. Studies have reported that *C. cassia* exhibits superior reduction in lipid peroxidation induced by FeCl2-ascrobic acid compared to α-tocopherol and possesses potent anti-superoxide formation activity [38]. Notably, our results demonstrated the induction of HO-1 and HSP90 in the *C. cassia*-administered group. HO-1 is primarily regarded as an antioxidant and has demonstrated anti-inflammatory properties in studies spanning the past two decades [39]. HSPs play a pivotal role in safeguarding the gastric mucosa, contributing significantly to mucosal defense mechanisms and ulcer healing by regulating the proteins associated with gastric cytoprotection and epithelial repair [40]. Moreover, the principal protective constituents of the gastric mucosa include glycoproteins such as mucin, prostaglandins, phospholipids, and peptide growth factors [41]. Notably, MUC5 and MUC6 are recognized for their roles in providing structural protection, thereby supporting the preservation of gastric mucosa [42]. In the current study, we postulated that the administration of high concentrations of *C. cassia* contributes to mucosal recovery by promoting the expression of MUC5 and MUC6, which helps to restore mucosal integrity disrupted by ethanol. 

Nevertheless, it is important to acknowledge the limitations of our study in elucidating underlying mechanisms. Further comprehensive research is necessary to delve deeper into these aspects. Furthermore, based on the aforementioned research findings, it can be cautiously inferred that *C. cassia* may not only be effective in gastritis but also in other digestive diseases such as inflammatory bowel disease stemming from inflammation. Therefore, further in-depth investigation is warranted to expand the knowledge about *C. cassia*’s potential therapeutic properties. 

## 5. Conclusions

Thus, *C. cassia* may be effective in mitigating ethanol-induced gastritis through the modulation of inflammatory signaling cascades. This study can serve as a significant foundation for the development of health-promoting dietary supplements utilizing non-toxic natural extracts, with potential applications in the prevention and treatment of gastric damage and gastritis arising from various etiologies, including ethanol.

## Figures and Tables

**Figure 1 nutrients-16-00055-f001:**
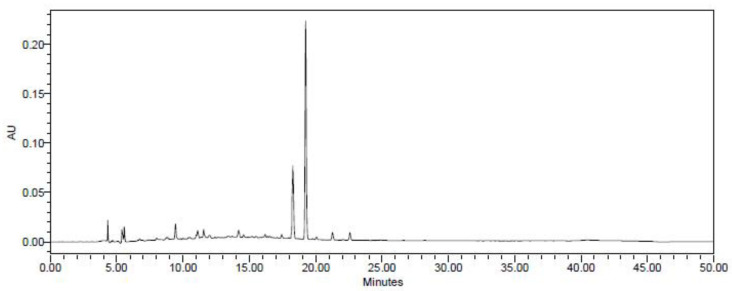
HPLC chromatogram of the *C. cassia* extract. Arrows show the peaks of cinnamic acid.

**Figure 2 nutrients-16-00055-f002:**
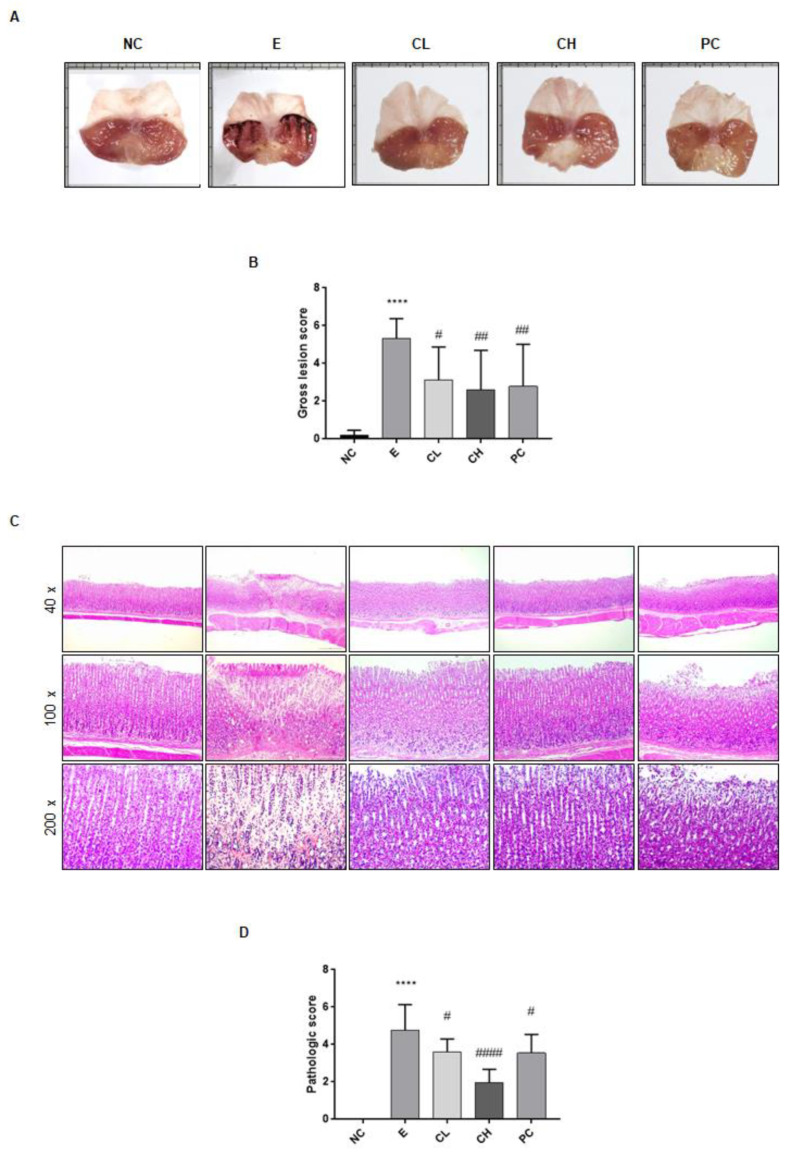
Effects of *C. cassia* on the gastric damage in ethanol-induced gastritis in SD rats. (**A**) Gross lesions in ethanol-induced gastric injury model. Representative images of gross lesions in each group. (**B**) The gross lesion scores. (**C**) Pathological changes were examined by H&E staining. (**D**) The pathologic score. Values are presented as mean ± SD (*n* = 9~10). Data were analyzed by Tukey’s test (**** *p* < 0.0001 vs. NC; #### *p* < 0.0001, ## *p* < 0.01, # *p* < 0.05 vs. E).

**Figure 3 nutrients-16-00055-f003:**
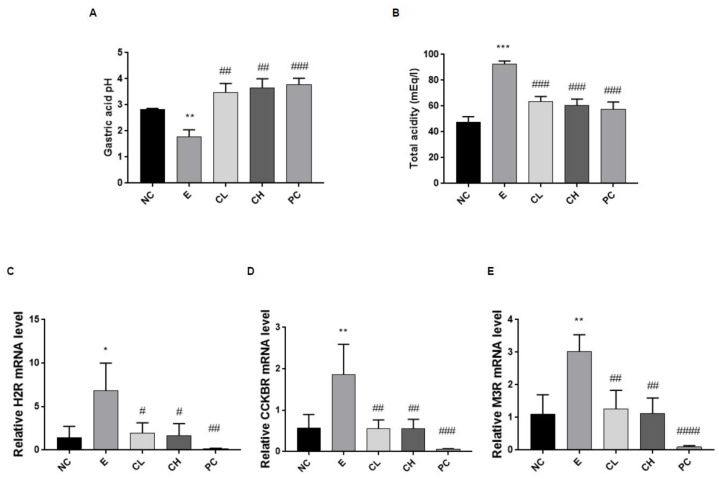
Effects of *C. cassia* on gastric acid secretion and the expression of gastric acid secretion-related receptors in gastric damage in ethanol-induced gastritis in SD rats. (**A**) Gastric acid pH and (**B**) total acidity of gastric juice in ethanol-induced gastric injury model. The mRNA levels of (**C**) *H2R*, (**D**) *CCK2R*, and (**E**) *M3R*. Value are presented as mean ± SD (*n* = 3~5). Data were analyzed by Tukey’s test (*** *p* < 0.001, ** *p* < 0.01, * *p* < 0.05 vs. NC; #### *p* < 0.0001, ### *p* < 0.001, ## *p* < 0.01, # *p* < 0.05 vs. E).

**Figure 4 nutrients-16-00055-f004:**
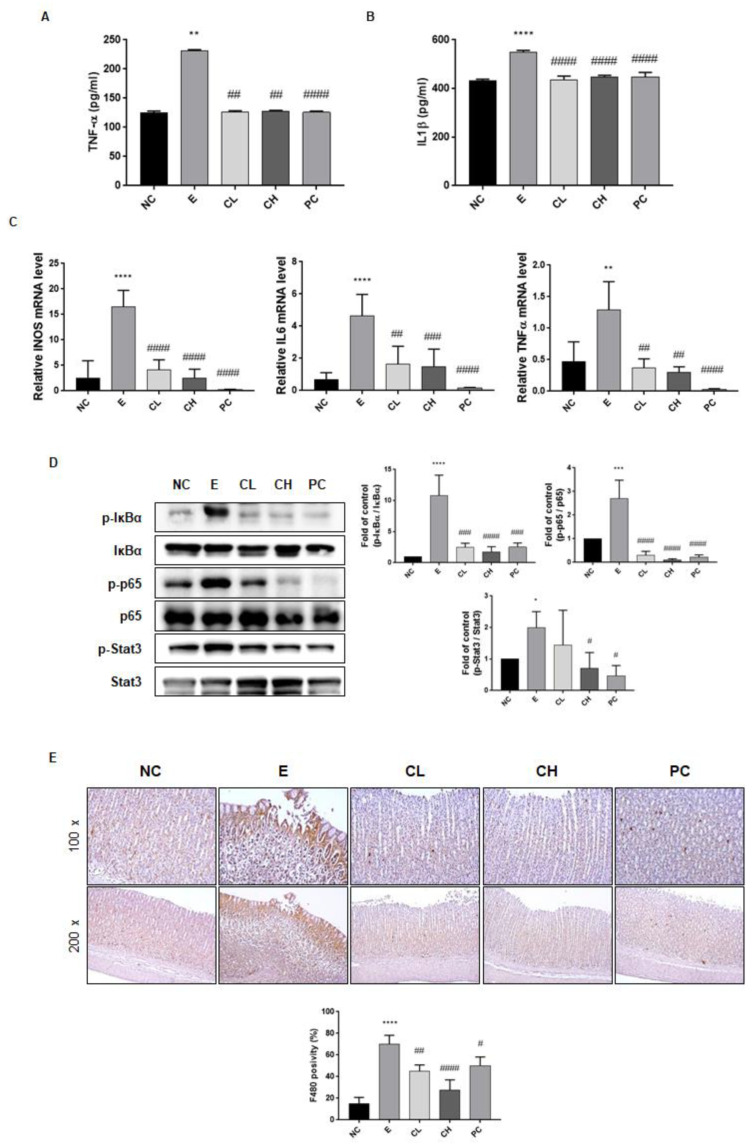
Effects of *C. cassia* on the inflammatory factors in gastric damage in ethanol-induced gastritis in SD rats. The expression levels of (**A**) TNF-α, and (**B**) IL-1β in serum were measured by ELISA. (**C**) The mRNA levles of *iNOS*, *IL-6*, and *TNF-α* in gastric tissue were analyzed by qRT-PCR. (**D**) The protein levels of p-IκBα, IκBα, p-p65, p65, p-stat3, and stat3 in gastric tissue were determined by immunoblotting. (**E**) The IHC changes in F4/80 staining, indicating macrophage infiltrations, were examined across the groups at two magnifications, ×100 and ×200. Values are presented as mean ± SD (*n* = 3~5). Data were analyzed by Tukey’s test (**** *p* < 0.0001, *** *p* < 0.001, ** *p* < 0.01, * *p* < 0.05 vs. NC; #### *p* < 0.0001, ### *p* < 0.001, ## *p* < 0.01, # *p* < 0.05 vs. E).

**Figure 5 nutrients-16-00055-f005:**
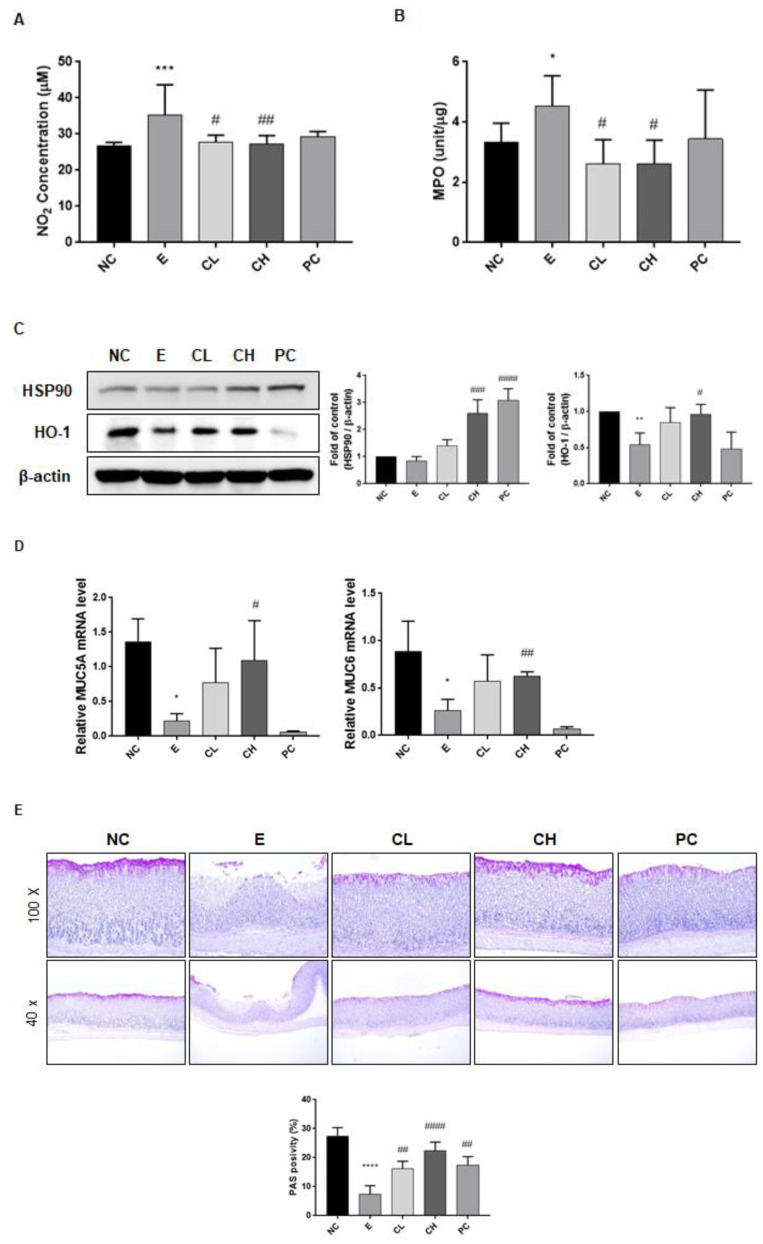
Effects of *C. cassia* on the antioxidant and defense mechanisms in gastric damage in ethanol-induced gastritis in SD rats. The expression levels of (**A**) NO and (**B**) MPO concentration in serum were measured by ELISA. (**C**) The protein levels of HSP 90, HO-1, and β-actin (as loading control) in gastric tissue were analyzed by immunoblotting. (**D**) The mRNA levels of *MUC5A* and *MUC6* in gastric tissue were analyzed by qRT-PCR. (**E**) Histological examination of the mucous layer was conducted using PAS staining, and this assessment was carried out in the various groups at two magnifications, ×40 and ×100. Values are presented as mean ± SD (*n* = 3–5). Data were analyzed by Tukey’s test (**** *p* < 0.0001, *** *p* < 0.001, ** *p* < 0.01, * *p* < 0.05 vs. NC; #### *p* < 0.0001, ### *p* < 0.001, ## *p* < 0.01, # *p* < 0.05 vs. E).

**Table 1 nutrients-16-00055-t001:** Groups for animal experiment.

Group	Route	Inducer	Tested Substance	Dose
NC	Oral	-	Saline	-
E	Oral	Ethanol	Saline	-
CL	Oral	Ethanol	*C. cassia*	20 mg/kg/day
CH	Oral	Ethanol	*C. cassia*	30 mg/kg/day
PC	Oral	Ethanol	Rebamipide	30 mg/kg/day

**Table 2 nutrients-16-00055-t002:** Antibodies for Immunoblotting.

Name	Cat. No.	Company
p-IkBα	#2859	Cell Signaling Technology (Danvers, MA, USA)
IkBα	#9242	Cell Signaling Technology (Danvers, MA, USA)
p-p65	#3033	Cell Signaling Technology (Danvers, MA, USA)
p65	#8242	Cell Signaling Technology (Danvers, MA, USA)
p- STAT3	sc-8001-R	Santa Cruz Biotechnology (Dallas, TX, USA)
STAT3	sc-483	Santa Cruz Biotechnology (Dallas, TX, USA)
HSP90	#4874	Cell Signaling Technology (Danvers, MA, USA)
HSP27	#50353	Cell Signaling Technology (Danvers, MA, USA)
HO1	ADI-SPA-895-F	Enzo Life Sciences (Farmingdale, NY, USA)
β-actin	sc-47778	Santa Cruz Biotechnology (Dallas, TX, USA)

**Table 3 nutrients-16-00055-t003:** Primer Sequences For qRT-PCR.

Gene	Primer Sequence	
*18S rRNA*	Forward	GCAATTATTCCCCATGAACG
	Reverse	GGCCTCACTAAACCATCCAA
*H2R*	Forward	CCATCCTGTACGCTGCTCTCA
	Reverse	TGCGAACTTGCAGTGGAAGA
*CCK2R*	Forward	GCGGAAACGTGCTCATCAT
	Reverse	GGCGTTGGTGACCGTTCTT
*M3R*	Forward	GCCTGGGTCTCTTAATTCCTATCA
	Reverse	ATGGGATCTGGATGGACACTTT
*iNOS*	Forward	GAGAAGCTGAGGCCCAGG
	Reverse	ACCTTCCGCATTAGCACAGA
*IL-6*	Forward	TCCTACCCCAACTTCCAATGCTC
	Reverse	TTGGATGGTCTTGGTCCTTAGCC
*TNF-α*	Forward	ACTGAACTTCGGGGTGATCG
	Reverse	GCTTGGTGGTTTGCTACGAC
*MUC5A*	Forward	TACCCCGAGCGTAGTGTACC
	Reverse	CAGGGGTCTTCACAGACGA
*MUC6*	Forward	ACCAGCCAAGTGACATCAACC
	Reverse	TGACCATGACTGATGCGTGG

18S rRNA, 18S ribosomal RNA; H2R, Histamine receptor 2; CCK2R, Cholecystokinin 2 receptor; M3R, Muscarinic receptor 3; iNOS, Inducible nitric oxide synthase; IL-6, Intereukin-6; TNF-α, Tumor necrosis factor alpha; MUC5A, Mucin 5AC; MUC6, Mucin 6.

## Data Availability

The data presented in this study are available on request from the corresponding author.

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
