# Peer review of "Gastric Mucosal Protective Effects of Cinnamomum cassia in a Rat Model of Ethanol-Induced Gastric Injury"

_nutrients, 2023, doi:10.3390/nu16010055_

Round 1

Reviewer 1 Report

Comments and Suggestions for Authors

The manuscript is interesting. It is worth to emphasizing that authors used a positive control with rebamipide, increasing the value of article. Nevertheless, some issues require revision. 

The lines 31 and 32 - if treated - in my opinion, if it remains untreated would be better sentence.

In the line 35, there are two dots.

In the initial part of the last paragraph of the Introduction section, you should emphasize that actually, plant-derived phytochemicals are novel agents for prevention or treatment of diseases of digestive tract, including gastritis (it is worth to emphasizing that there is a need not only gastritis, and also other diseases of digestive tract diseases). This data should be supported by some most recent articles, among other doi: 10.1016/j.biopha.2019.109594, 10.1080/87559129.2023.2273929, 10.1016/j.biopha.2017.07.007.

In the last paragraph of the Introduction section, the description being characteristics of Cinnamomum cassia is very poor. You should mention about main compounds contained in this plant, as well as their chemical nature, their detailed properties.

In the last paragraph of the Introduction section, you should directly emphasize what is advantage of your work over other articles assessing Cinnamomum cassia in gastric mucosal damage. What is the novelity?

What is solvent for dry extract? It should be included in the section of materials and methods.

In the discussion section, there is a lack of comparing your results with results of other team assessing Cinnamomum cassia or cinammic acid in models of gastric diseases.

The creation of a figure representing effects of activity of Cinnamomum cassia in gastritis would contribute to a better representation of the manuscript.

Reviewer 2 Report

Comments and Suggestions for Authors

In this paper, the author investigated the potential gastroprotective property of C. cassia and the underlying mechanisms of action in a rat model of ethanol-induced gastric injury. The results indicated that the rat model pretreatment with C. cassia led to decreased levels of inflammatory factors, including TNF-α, p-p65, and IκBα. The C. cassia not only upregulated the expressions of HO1 and HSP90, but also enhanced expression of PAS and MUC. In summary, the logic of this paper is clear and reasonable, but there are still some problems as follows.

1. There is much volatile oil in C. cassia. Why did the author just use HPLC to determine cinnamic acid without discerning volatile oil part?

2. The author can specify the specific side effects of prescription drugs for gastric ulcer (proton pump inhibitor PPI and h2r antagonists) in the introduction section, to highlight the necessity of using safer natural products without side effects to develop methods for the treatment of gastritis and gastric ulcer.

3.  In 2.3 Animal Experiments section, it can be inserted a table to show the grouping in detail.

4. More magnification times of HE should be applied instead of Figure 2B (40 times). In addition, measure should be added in each HE figures.

5. What makes C. cassia safer than rebamipide? The results section mainly focused on both C. cassia and rebamipide can improve gastric injury (except 3.4) without comparing C. cassia with rebamipide.

6. The figures are arranged slightly closely, and it is suggested that the author making appropriate adjustments.

7. Most of the references are old. It is recommended referring to articles of recent 5 years.

Reviewer 3 Report

Comments and Suggestions for Authors

Round 2

Reviewer 1 Report

Comments and Suggestions for Authors

Authors revised manuscript according to provided comments and suggestions

Reviewer 3 Report

Comments and Suggestions for Authors

The authors have addressed most of the comments. So, the paper can be accepted.